# An Overview of Healthcare Associated Infections and Their Detection Methods Caused by Pathogen Bacteria in Romania and Europe

**DOI:** 10.3390/jcm11113204

**Published:** 2022-06-04

**Authors:** Sándor Szabó, Bogdan Feier, Denisa Capatina, Mihaela Tertis, Cecilia Cristea, Adina Popa

**Affiliations:** 1Department of Analytical Chemistry, Faculty of Pharmacy, “Iuliu Hațieganu” University of Medicine and Pharmacy Cluj-Napoca, 400349 Cluj-Napoca, Romania; szabo_sandor88@yahoo.com (S.S.); feier.george@umfcluj.ro (B.F.); denisa.elen.capatina@elearn.umfcluj.ro (D.C.); mihaela.tertis@umfcluj.ro (M.T.); 2Department of Clinical Pharmacy, Faculty of Pharmacy, “Iuliu Hațieganu” University of Medicine and Pharmacy Cluj-Napoca, 400349 Cluj-Napoca, Romania; apopa@umfcluj.ro

**Keywords:** healthcare-associated infections, Romania, bacteria, antibiotics, bacteria detection, infection diagnosis methods

## Abstract

Healthcare-associated infections can occur in different care units and can affect both patients and healthcare professionals. Bacteria represent the most common cause of nosocomial infections and, due to the excessive and irrational use of antibiotics, resistant organisms have appeared. The most important healthcare-associated infections are central line-associated bloodstream infections, catheter-associated urinary tract infections, surgical site, soft tissue infections, ventilator-associated pneumonia, hospital acquired pneumonia, and *Clostridioides difficile* colitis. In Europe, some hospitalized patients develop nosocomial infections that lead to increased costs and prolonged hospitalizations. Healthcare-associated infection prevalence in developed countries is lower than in low-income and middle-income countries such as Romania, an Eastern European country, where several factors contribute to the occurrence of many nosocomial infections, but official data show a low reporting rate. For the rapid identification of bacteria that can cause these infections, fast, sensitive, and specific methods are needed, and they should be cost-effective. Therefore, this review focuses on the current situation regarding healthcare-associated infections in Europe and Romania, with discussions regarding the causes and possible solutions. As a possible weapon in the fight against the healthcare-associated infections, the diagnosis methods and tests used to determine the bacteria involved in healthcare-associated infections are evaluated.

## 1. Introduction

Healthcare-associated infections (HAI), also called nosocomial infections, are infections acquired during the process of receiving health care that were not present at the time of admission. They can occur in different care units and can affect both patients and healthcare professionals [1]. Worldwide, 5–10% of patients develop HAI [2]. The most important HAI are central line-associated bloodstream infections (CLABSI), catheter-associated urinary tract infections (CAUTI), skin and soft tissue infections (SSTI), surgical site infections (SSI), ventilator-associated pneumonia (VAP), hospital acquired pneumonia (HAP), and *Clostridioides difficile* colitis (CDI), with bacteria causing about 90% of HAI [1]. Due to excessive use of antibiotics, almost 20% of all reported bacteria are multidrug-resistant (MDR) and are among the major complications of HAI [3].

In Europe, it is estimated that around 80,000 hospitalized patients have at least one HAI on any given day, and this leads to 16 million additional hospitalization days each year [4]. HAI prevalence in high-income countries is around 7.5%, while in low- and middle-income countries, the prevalence rate ranges between 5.7% and 19.2% [4,5].

In Romania, HAI represent a much-underestimated pathology, with the official prevalence rates of only 0.2–0.25%, due to many factors that contribute to the underreporting of HAI [4]. With the approval of Order no. 1101 from 2016, the reporting rate of HAI has increased, but the prevalence rate is still far from the European average of 7.1% [6,7]. The most problematic HAI are CDI and CRE infections, which are growing alarmingly and contribute to the high costs of the health system [8,9,10]. It is necessary to implement strategies for improving the activities of surveillance, prevention, and limitation of infections in medical units and reducing the causes of under-reporting of HAI [11].

HAI lead to increased direct and indirect costs, with a wide variation in costs between hospitals and countries. In the USA, the costs of HAI can reach ~$10–33 billion per year, and in the EU HAI generate costs exceeding 7 billion € annually [4,12,13]. CLABSI was found to be the costliest HAI, at a cost of $31,000–65,000 per episode [14,15]. Control and prevention strategies have been shown to be effective and efficient [16,17,18,19] and the prevention programs should take into consideration the local situation of the healthcare setting.

One major cause for the spread of HAI is the limited capacity to diagnose these infections. For the rapid identification of bacteria that can cause HAI, fast, sensitive, and specific methods are needed, and finally they should be cost effective. The conventional approach used to detect and identify bacteria is based on traditional culture methods, but these have limitations, so recently there has been a huge interest in the development of rapid detection methods, which have demonstrated their usefulness in clinical practice [20,21,22,23,24]. This review presents the main aspects related to HAI (the rates, types, causes, identification, and treatment of HAI), painting a general picture of HAI worldwide in order to better understand the HAI aspects in Romania, where the data are limited, with low official prevalence rates and with few and regional studies only. The review presents the health and economical arguments, with worldwide examples, building the case for the better application of HAI surveillance. To the best of our knowledge, there is no review that presents in such detail the issue of HAI, focusing on the situation of HAI in Europe and Romania. In addition, the present review centralizes and evaluates the analytical methods used for the detection of bacteria causing HAI.

## 2. Healthcare Associated Infections

The HAI first appear 48 h or more after hospital admission [25,26]. HAI may occur in different areas of healthcare settings, such as in hospitals, long-term care facilities (LTCF), and ambulatory settings, and may appear after discharge, within 30 days [1,25]. Any infection is considered to be nosocomial if a patient was hospitalized in an acute care hospital (ACH) for two or more days within 90 days of the infection, or resided in a nursing home or LTCF, or received recent intravenous therapy, or wound care within the past 30 days of the current infection [27]. HAI may also include occupational infections that affect staff in healthcare settings. HAI are not related to the disease for which the patient is hospitalized but occur in care units. The number of nosocomial infections seems to be increasing for various reasons, for example, hospitals caring for an increasing number of patients, increasing antibiotic resistance, transferring pathogens from medical staff to the patient or from the environment to the patient, non-compliance or lack of sanitation protocols, too little emphasis on prevention. HAI affects 3.2% of all hospitalized patients in the United States of America (USA), 6.5% in the European Union/European Economic Area (EU/EEA) [1], and worldwide, approximately 10% of patients acquire a HAI, resulting in prolongation of the hospital stay, increase in cost of care, and significant morbidity and mortality [28]. Pathogens responsible for nosocomial infections include bacteria, viruses, and fungi. The prevalence of infections caused by microorganisms varies depending on the healthcare facility location, healthcare setting, and patient population.

Due to excessive and irresponsible use of antibiotics, resistant bacteria have appeared. These multidrug resistant (MDR) bacteria are one of the complications of HAI. Studies have shown that almost 20% of all reported bacteria are MDR [3]. Vancomycin-resistant *Enterococci* (VRE) and methicillin-resistant *Staphylococcus aureus* (MRSA) are the major Gram-positive pathogens of concern, while *Pseudomonas aeruginosa*, *Klebsiella pneumoniae*, and *Enterobacter* spp. are the major resistant Gram-negative pathogens [25,29,30,31,32,33]. For MRSA the major mode of transmission is from the contaminated hands of healthcare workers to the patients [34]. MRSA currently accounts for more than 50% of *S. aureus* strains isolated from hospital patients in the USA and causes approximately 50% of all nosocomial *S. aureus* infections [30]. The SCOPE (Surveillance and Control of Pathogens of Epidemiologic Importance) project found that Gram-positive bacteria have highly variable growth and resistance patterns and accounted for 62% of all CLABSI in 1995 and 76% in 2000, in an ascending manner [35]. Resistance to carbapenems has been found in many Gram-negative species, including both *Enterobacteriaceae* (e.g., *Escherichia coli*, *Enterobacter* spp., *Serratia* spp.) called Carbapenem-resistant *Enterobacteriaceae* (CRE) and non-fermenters (e.g., *P. aeruginosa* and *Acinetobacter baumannii*), but *K. pneumoniae* is the most frequent species [36]. A meta-analysis evaluating the number of deaths attributable to CRE infections found that 26–44% of deaths were attributable to carbapenem resistance [37]. Gram-negative bacteria caused 22% of CLABSI in 1995 and 14% in 2000 [35]. These infections can be complicated by CDI [28]. Table 1 broadly presents the characteristics of these different categories of HAI.

The mortality rate differs depending on the healthcare units in which these data were observed and depends very much on the degree of compliance with the guidelines for reducing these complications of HAI. At the same time, it is observed that these HAI can be prevented, but there are several factors that can change the final percentage, such as: the degree of compliance with protocols, compliance of medical staff, complications, or the performance of the health system.

According to the recommendations [54] of the International Society for Infectious Diseases, hospitals are encouraged to implement a multidisciplinary team to manage the use of antibiotics through the antibiotic stewardship program. An infectious disease physician and a clinical pharmacist with infectious disease training form the core team. Prevention is the base of the management of HAI, but when antimicrobial treatment is necessary, there are three groups of antimicrobials that can be used in the management of infections [55] (as shown in Table 2) in accordance with the results of the microbiological cultures and the antibiogram and the curative treatment.

The most common HAI in hospital settings is CAUTI, followed by VAP, SSI, CLABSI and CDI. In acute hospital settings, the most frequent HAI is pneumonia, followed by gastrointestinal infections, SSI, and UTI [1]. MRSA is the most common MDR organism in USA hospitals and, according to recent estimates, is responsible for over 50% of CLABSI, almost 60% of CAUTI, almost 50% of VAP, and just over 40% of SSI [43]. In EU/EEA, the situation seems to be different. The most frequently isolated microorganism causing HAI is *E. coli* (16.1%), followed by *S. aureus* (11.6%), *Klebsiella* spp. (10.4%), *Enterococcus* spp. (9.7%), *P. aeruginosa* (8.0%), *C. difficile* (7.3%), coagulase-negative staphylococci (7.1%), *Enterobacter* spp. (4.4%) and *Proteus* spp. (3.8%) [29]. In a prospective observational incidence study, conducted in Scotland in 2018, the results showed that the most common organisms causing HAI were Gram-negative-bacilli, being responsible for 51% of CLABSI and 75% of UTI, with *E. coli* being the most frequently isolated bacterial species (18.4%) [56]. In the USA, approximately 2 million infections and 23,000 deaths are caused by MDR organisms [13], with a total cost of antimicrobial resistance (AMR) up to US$ 20 billion per year [57], while in Europe, more than 3 million patients acquire HAIs every year, with 37,000 deaths occurring as a direct consequence of these infections associated with the increasing AMR of the incriminated pathogens [58]. HAI, particularly those acquired in a critical care setting contribute significantly to morbidity and mortality, and the supplementary health care costs [2,13] are a financial burden on the healthcare systems [1]. One in 20 hospitalized patients acquires a HAI while receiving medical care [2]. HAI affect as many as 1.7 million patients [13] at a cost of ~$10–33 billion and up to 99,000 lives [12,13] in USA hospitals annually and in Europe, it is estimated that 6.5% of patients treated in an ACHhave an HAI [56]. Patients with HAI have a larger proportion of readmissions compared with patients with no HAI and generate excess costs for the healthcare system [59]. Recent analyses show that at least 50% of HAI are preventable [60], indicating that prevention could lower the economic burden of HAI. A WHO report found that the SSI are the most investigated, and hence the most frequent, HAI in low-income and middle-income countries, with more than 10% (up to 30%) of operated patients usually developing SSI and *S. aureus* as the most frequent cause of SSI. In these countries, the level of risk for patients undergoing surgical procedures is significantly higher than in developed countries, where SSI rates vary between 1.2% and 5.2%. SSI can prolong hospital stay up to 21 days in settings with limited resources, adding a burden on the patient suffering and the financial cost [61]. Excess length of hospital stays (LOS) caused by HAI and estimated costs of HAI worldwide are shown in Table 3.

In the USA, CLABSI was found to be the costliest HAI, witha median cost of $45,814. CLABSI caused by MRSA resulted in even higher associated costs to hospitals, up to $58,614. CLABSI and SSI caused by MRSA resulted in the highest attributable excess LOS (15.7 and 23.0 days, respectively) [15]. In total, annual costs for the major HAI in the USA could reach more than $10 billion [1,15]. In the USA, the annual additional costs of infections caused by AMR bacteria are estimated between $21 billion and $45 billion [70,71]. Studies have shown that implementing infection prevention and control programs can reduce LOS and avoid additional costs. It is estimated that hospitals can avoid between 12,000 to 223,000 HAI and save $142 million to $4.25 billion annually with infection prevention measures [72]. These prevention measures, according to the Romanian healthcare regulations [7] and the WHO recommendations [18], should include hand hygiene using alcoholic solutions or soap and water, use of personal protective equipment, using aseptic and safe practices for injecting, preparing and administering parenteral medicinal products, safe handling of medical equipment or contact with potentially contaminated surfaces, decontamination of medical devices and patient care equipment, respiratory hygiene and cough management, environmental cleaning, healthcare waste management, triage of infectious patients, and basic principles of standard and transmission-based precautions. Eliminating one HAI could cost around $25,000 but it can generate around $582,000 profit, so reduction of HAI could be profitable to hospitals [73]. Unfortunately, the economic impact of HAI in low-income and middle-income countries is poorly studied.

### 2.1. HAI in Europe

Within the EU it is estimated that annually over 4 million patients acquire a nosocomial infection, of whom approximately 37,000 die as a result [74].

In Europe, the most common HAI are caused by around ten bacterial species (Figure 1 [4]).

In Europe, HAI results in costs exceeding €7 billion annually [4]. For example, a study showed that in Germany, HAI can generate additional costs of €5823–€11,840 ($7453–$15,155) per infected patient [75].

The incidence of different types of HAI varies from country to country depending on the development of the medical system. In Europe, it is estimated that around 80,000 hospitalized patients have at least one HAI on any given day, and HAI causes 16 million additional hospitalization days each year [4]. HAI prevalence in high-income countries is up to 7.5% [4,5], although others have reported rates of 5.7–7.1%, while in low-income and middle-income countries, the prevalence rate ranges between 5.7% and 19.2% [4,5]. According to the World Bank, high-income countries have a gross national income (GNI) per capita of at least $12,476, upper-middle-income countries have the GNI per capita between $4038 and $12,475, lower-middle-income countries have the GNI per capita of $1026 to $4035 and low-income countries have the GNI per capita of $1025 or less [76].

Among European hospitals, the prevalence of at least one HAI varied based on the care setting: 4.4% in primary care hospitals. 7.1% in tertiary care hospitals, 19.2% in ICU; and 3.7% in LTCF [29]. Approximately 8.9 million distinct HAI episodes occur annually in ACH and LTCF within the EU, including 4.5 million in ACH and 4.4 million in LTCF [1,29].

Point prevalence surveys of HAI from 2016 to 2017, including 28 countries from Europe, showed that 6.5% of patients in ACH and 3.9% of patients in LTCF had at least one HAI and the total number of residents with at least one HAI on any given day in LCTF was estimated at 129,940 [29]. The most frequently reported type of HAI in ACH and LTCF was respiratory tract infections, with a proportion of 21.4–33.2% [29]. In ACH, the infections responsible for HAI were UTI (18.9%), SSI (18.4%), CLABSI (10.8%) and gastro-intestinal infections (8.9%), with CDI accounting for 44.6% of the latter and 4.9% of all HAI [29]. Thirty percent of patients in ICU have at least one HAI. In low- and medium-developed countries, the frequency of HAI in the ICU is at least two to three times higher than in developed countries, and the frequency of infections associated with devices is thirteen times higher than in the USA [5]. According to a European report [77], the percentage of pneumonia or lower respiratory tract infections ranged from 12.0% in Sweden to 36.3% in Lithuania. The percentage of lower tract infections ranged from 10.1% in Cyprus to 30.7% in France. The proportion of SSI ranged from 8.8% in Luxembourg to 29.0% in Spain. The rates of CLABSI were highest in Greece (18.9%) and Cyprus (19.0%) and the lowest in Iceland (2.0%) and were secondary to another infection in 28.8% of cases [4]. HAI associated with VAP shows a declining tendency in countries with well-developed medical systems, e.g., in Germany from 11.2 to 8% [78] and in France from 14.7 to 12.6% [79]. In the case of CAUTI, the reported incidence was 3.43 events per 1000 patient-days in EU countries [80]. The reported incidence of CLABSI across 422 ICU in 36 countries in Latin America, Asia, Africa, and Europe from 2004 to 2009 was 6.8 events per 1000 central-line days [81].

AMR observed in HAI was 31.6% in ACH and 28.0% in LTCF [29]. In a recent publication, more than 425.000 HAI caused by AMR microorganisms occur in the EU every year [82]. Attributable deaths in the EU due to AMR bacteria were estimated to be 33.110 per year [82]. In addition to MRSA, MDR *E. coli* [82] and CRE are becoming an important problem for public health [83]. *Enterococcus* spp., especially *E. faecium* and *E. faecalis* have also received particular attention due to their ability to acquire MDR against many antimicrobial agents used in clinical practice and establish life-threatening infections in patients living with cancer or chronic diseases. In a 5-year study conducted in Salerno, Italy, *E. faecium* showed high resistance rates against imipenem (86.7%), ampicillin (84.5%), and ampicillin/sulbactam (82.7%), while *E. faecalis* showed the highest resistance rate against streptomycin (67.7 %) and gentamicin (59.3%) [84]. Millions of antibiotic prescriptions are prescribed to patients each year, but it is estimated that approximately 50% of these are not necessary [85]. Pharmacists and clinical pharmacists may be involved through the monitoring of antimicrobial stewardship programs to limit inappropriate antibiotic use and help to prevent the spread of resistant pathogens [1,86]. Stewardship programs can achieve significant cost savings, particularly drug cost savings [87,88].

### 2.2. HAI in Romania

In Romania, HAI represent a much-underestimated pathology [4]. According to the official reports of HAI communicated by hospitals, the prevalence rates are only of 0.2–0.25% [4]. More accurate data for Romania were identified in a European study in 2018, which showed a prevalence of 2.6%, which would represent approx. 100,000 cases registered annually in Romania [4].

One of the identified causes of the high number of HAI is the architecture of hospitals, as these hospitals were built many years ago and most of them do not meet current space requirements. The wards are small or crowded, with several beds, so the risk of transmitting pathogens increases, or there are no designated spaces for the patient’s isolation [4]. Other causes that contribute to the growth of HAI are MDR bacteria, specifically CRE, with 68.9% resistant isolates contributing to more than 13,000 cases of HAI in ACH [29]. However, there are methodologies for surveillance of HAI in the sentinel system and microbial resistance elaborated in Romania to ensure the standardization of case definitions, data collection and reporting of data collected at the sentinel units, to improve the quality of patient care [89].

A deadly fire occurred at Colectiv nightclub, in Bucharest, Romania, on the night of 30th of October 2015, leading to the death of 64 people (26 on site, 38 in hospitals) and the injury of 144 people [90,91]. This event has had an important impact on Romanian society. The collective outpouring of grief over the loss of life led to memorial gatherings, which transformed into large-scale, nationwide protests that lasted for nine days and forced the Government to resign [92]. These protests revealed a form of societal resilience. The stronger than previously thought post-communist civil society challenged rooted issues of corruption and inequality, as well as the institutions and people enabling them [92].

The deaths in hospitals of people injured in the fire raised awareness about HAI in Romania [90,91,93]. Burn wounds are highly prone to long-term colonization by nosocomial bacteria, making their treatment more difficult. Romania is a country with a high prevalence of carbapenemase-producing microorganisms [94], which led to complicated infections and even death in the Colectiv patients [91]. After an initial denial, the Romanian Government admitted that appropriate medical care could not be provided for all the patients from the fire, and about 80 patients hospitalized in Romania for more than a week were transported to hospitals in Western Europe. For example, in the case of a patient transported to the Netherlands [91] five different carbapenemase-producing isolates of *K. pneumoniae*, *A. baumanii*, *Enterobacter cloacae*, *Providencia stuartii* and MDR *P. aeruginosa* were identified. A second wave of the scandal was triggered in the spring of 2016 when a team of investigative journalists discovered a chain of fraud related to disinfectants used in the healthcare system, as a possible reason for the high number of HAI in Romania [90,93]. All these factors led the Government to adopt legislation for combating the HAI [7].

With the approval of Order no. 1101 from 2016 [7], regarding the approval of the Norms of surveillance, prevention, and limitation of the HAI in the healthcare units, in Romania the reporting rate of HAI has increased, but the prevalence rate is still far from the European average of 7.5% [6]. In Order no. 1101/2016, HAI has a definition that emphasizes that these infections are contracted in sanitary units with beds (state and private-owned), which affects either the patient due to the medical care received, or the healthcare staff members due to their activity, and it must be proved that the infection is due to the hospitalization or the medical-sanitary care in the sanitary units [7]. The objective of the order is to increase the quality of medical services and patient safety by reducing the risk of HAI [5,7].

In Romania, the identification of suspected cases of HAI is done by the attending physician and the case definitions of HAI provided in Decision 2012/506/EU should be respected [95]. One study followed the HAI rate for a 2-year period (2017–2018) in a county of about 580,000 inhabitants, in which 1024 cases were reported from seven public hospitals. The most frequent HAI were reported in the ICU (48.4%) and the most frequent infections were: bronchopneumonia (25.3%), followed by CDI (23.3%). Of the reported cases, 25.3% were declared deaths related to *A. baumannii* (39.2%) and *P. aeruginosa* (32.2%) infections [4]. The five most common bacteria were *C. difficile*, *A. baumannii*, *K. pneumoniae*, *P. aeruginosa* and *S. aureus* [4]. Most of the bronchopneumonias had, as etiology, infections with *A. baumannii* (60.8%), surgical wound infections with *S. aureus* (41.8%), UTI with *E. coli* (29.2%) and *K. pneumoniae* (28.9%), and CLABSI caused by *K. pneumoniae* (28.3%) and *S. aureus* (23.6%) [4].

The most problematic HAI in Romania is CDI, which has been increasing since 2011 and in one Romanian hospital the number of cases has increased since 2011 sevenfold, and twenty-two times in 2012 compared to 2010 [8], and is the highest proportion recorded in the EU, at about 70% of all cases [9], with huge costs for the healthcare system [8]. The second most problematic HAI in Romania is Carbapenemase-producing *Enterobacteriaceae* infections. The EARS-Net report for 2012–2013 showed extremely high levels of resistance: in the case of *P. aeruginosa*: 58–60% (first place in Europe), *A. baumannii*, 81–85% (second to third place in Europe) and *K. pneumoniae* 14–21% (third place in Europe) [96]. Some data show resistant isolates of 33.8% with an estimated HAI caused by CRE of 3475 cases in ACH [29]. The death rate from systemic infections caused by CRE reaches 40–50% [97,98].

According to the National Center for Statistics and Informatics in Public Health, the number of HAI reported out of all patients in 2017 was 19,607 cases (around 975 cases/1M population), of which most were digestive (8019 cases), respiratory (3549), urinary (2568) and infected surgical wounds (2297). The incidence of HAI in 2015 was 0.33%, and in 2016 was 0.44% of the total number of patients. After the implementation in September 2014 of the national system for surveillance of infections caused by CDI, the HAI reports have increased; thus, since 2014, respiratory infections have given way to digestive infections. For 2016, digestive infections accounted for 35.2% of the total HAI reported; their number increased by 39.8% compared to 2015 and by 402% compared to 2013 [5]. In 2018 the reported HAI incidence was only 0.55%. Between 2015 and 2018, 40.1% of the reported HAI cases had a digestive localization, and the number of reported HAI increased by 90.4% compared to 2014 [5]. The prevalence of CDI assessed in a multicenter study was 5.2 cases per 10,000 patients per day [99], and in 2019 the incidence was 0.63% [100]. Figure 2 presents a general view of the situation of HAI in Romania.

Even though the national rates of HAI for Romania are much lower than the European average, different regional, small studies paint a different picture for the HAI situation in Romania. ESKAPE pathogens (*E. faecium*, *S. aureus*, *K. pneumoniae*, *A. baumannii*, *P. aeruginosa*, *E. coli*) are present in Romanian hospitals. One retrospective study (2016–2020) [101] assessed the AMR of ESKAPE pathogens isolated from Romanian patients’ biological samples, which included 4293 bacterial isolates of which 67% had Gram-negative bacilli, 31% Gram-positive cocci and 2% other morphotinctorial bacteria. ESKAPE pathogens were found in 97% of the bacterial isolates strains, with *E. coli* (38.26%) and *S. aureus* (26%) the most prevalent ones. Increased AMR was observed for MRSA, ESBL, Enterobacterales, carbapenem-resistant *P. aeruginosa*, *A. baumannii* and *K. pneumoniae*. No vancomycin resistance was found for *E. faecium*. The highest prevalence rates of MDRwere found in MRSA (86.6%), *A. baumannii* (36.8%), *P. aeruginosa* (29.1%) and *Klebsiella pneumoniae* (24.4%) [101].

Another study on UTI in Romania revealed that 212 samples with Enterococcus showed a high resistance profile for levofloxacin and penicillin (32.07%), and ampicillin (14.62%) [102]. The European Centre for Disease Prevention and Control (ECDC) Surveillance Atlas on AMR reported the VRE proportion rates for Romania to be 39% [103].

From 2004 to 2005, 60–72% of invasive *S. aureus* isolates from Romanian hospitals were resistant to methicillin [104]. Clinical isolates from CLABSI, SSI, as well as from screening swabs, were collected at one hospital and nearly half of all isolates (47%), and about one third (34%) of bloodstream isolates presented MRSA [105]. In 2020, according to ECDC 47.3% of the isolates contained MRSA [106].

A retrospective study on 270 urine samples [102] showed that *K. pneumoniae* had a 28.62% resistance to amoxicillin-clavulanic acid, 15.61% to levofloxacin and nitrofurantoin, and 15.24% to ceftazidime. Another study [107] analyzed the antimicrobial susceptibility of *K. pneumoniae* strains isolated from blood in 2010 and 2015 from a Romanian hospital: 18 strains were identified in 2010 and 37 strains in 2015. Although the resistance to aminopenicillin-betalactamase inhibitors, piperacillin-tazobactam, third generation cephalosporins, fluoroquinolones, gentamicin, amikacin and combined resistance decreased between these two-time frames, this change was statistically non-significant. The same was noticed for the increased resistance rates to carbapenems. According to ECDC reports in 2020, *K. pneumoniae* had 67.9% resistance to third generation cephalosporins, 48.3% to carbapenems, and 66.2% to fluoroquinolones [106]. According to the same report, *A. baumanii* in 2020 had a 93.3% resistance to carbapenems, 95.3% to fluoroquinolones, and 90.1% to aminoglycosides [106].

The Atlas on AMR showed that in 2020 *P. aeruginosa* had a 42.1% resistance to piperaciline-tazobactam, 46.4% to fluoroquinolones, 41% to ceftazidime, 37.1% to aminoglycosides, 43.9% to carmapenems [106], which are lower rates than in 2018 [103].

In Romania, *E. coli* has developed resistance to multiple antibiotics, such as fluoroquinolones (30.4%), third generation cephalosporins (22%), aminoglycosides (19.6%), and a multi-resistance of 10.9% [108]. Another study [109] determined retrospectively the antibiotic susceptibility of *E. coli* strains isolated from a pediatric population hospitalized in a south-eastern infectious disease hospital. They found a low sensitivity to ampicillin (19.6%), tetracycline (29.5%), and amoxicillin (37.5%). The highest sensitivity was to carbapenems (93%). A retrospective study [102] observed the resistance profile of *E. coli* isolated from urine samples. Of 957 samples with *E. coli*, the resistance profile was 29.66% for levofloxacin, 14.13% for amoxicillin–clavulanic acid, and 6.68% for ceftazidime.

An activity report [110] in 2019 regarding HAI found that one of the causes of low incidence of HAI in Romania was represented by the non-conformities found in Romanian hospitals. Of the analyzed obstetrics and gynecology clinics, 25% did not comply with the annual nosocomial infection surveillance and control plan, in 8.3% of the units there was no register of nosocomial infections, in 8.3% of the units there were no data on the incidence and prevalence of nosocomial infections, and in 8.3% of units there was no isolation of patients with high-risk infections. In many clinics and private offices there was no record of HAI, even though there are national programs for surveillance and control of nosocomial infections that provide money to reduce the number of HAI.

Up to 70% of HAI can be prevented through effective infection prevention and control measures [16,17,18,19]. The WHO is continuously developing documents on HAI prevention [111,112]. Multifaceted HAI prevention programs are cost-effective [113] and leadership engagement and data-driven interventions with frequent performance feedback are also important facilitators of HAI prevention [114]. One of the methods by which HAIs can be reduced is the use of antibiotic stewardship programs. The purpose of ASP is to reduce the use of unnecessary or inappropriate antimicrobials in health care settings with the goal of slowing the rate of development of antibiotic resistance [86]. An important strategy to reduce AMR is the use of rapid diagnosis for bacterial infections. Studies have shown that the use of rapid diagnosis methods for bacterial infections as part of ASP can improve the time to optimal antibiotic therapy, decreasing the rate of recurrent infection, mortality, hospital duration and hospital costs [115].

## 3. Testing Methods for Diagnosis Purposes of Bacteria Involved in Nosocomial Infections

Bacterial infections are among the leading causes of HAI, and identifying the causative organism can be challenging, especially for resistant strains, requiring fast and accurate methods of detection. The requirement for bacteria typing is also necessary, given the observed emergence of diverse types of virulent strains. Deciding which test to implement for the correct and rapid identification of these pathogens, laboratories must consider the sensitivity and complexity of the method, the turnaround time, the expertise required for each test and the cost of the analysis. The conventional approach used to detect and identify bacteria is based on traditional culture methods, which are still the gold standard due to their reliability, efficiency, sensitivity, and range of applications [116,117]. However, culture methods are laborious and require long time for bacteria to grow, the results being reported in 1–5 days [118]. Because of these limitations, it is recommended that laboratories supplement or replace culture-based approaches with other detection methods. Recent advances in molecular and nonmolecular testing methods greatly reduce the time required to detect the bacteria involved in HAI [119]. Microbiological techniques for rapid diagnosis allow quick identification of bacteria, which is necessary for the early management of patients. Rapid diagnosis testing with ASP intervention has an 80% chance of being cost-effective, compared with testing without ASP intervention [20]. The diagnosis tools used for the detection of bacteria involved in nosocomial infections are categorized into nucleic acid-based, biosensor-based, immunological-based and mass spectrometry-based methods, as shown in Table 4.

Conventional methods for the detection of pathogenic bacteria are based on bacterial culture and involve several steps: sample preparation, enrichment, dilution, plating, enumeration, and isolation of single species colonies on selective media for further characterization [116]. Besides the long analysis time and laborious steps, the conventional methods have other limitations that make these methods inappropriate for field applications or situations that require immediate results: the need for special analysis conditions (temperature, light), low specificity compared to other methods, and the need for considerable quantities of consumables and qualified personnel [128]. Despite these limitations, classical methods are still the most widely used tests in routine laboratories to identify bacteria, including those that produce HAI. Chromogenic agars, which contain antibiotics that allow only the development of resistant bacteria, represent an adaptation of the traditional culture methods and are increasingly used for the detection of several bacteria in clinical laboratories. This new generation of media represents a sensitive, convenient, and relatively low-cost method for identifying the pathogens based on a color reaction produced in the bacterial culture, with a shorter turnaround time [117,125,128,129,130]. Some examples of commercially available chromogenic agars for bacteria involved in HAI are presented in Table 5.

In recent years, more powerful molecular, immunological, and biochemical analytical methods have emerged to overcome the limitation of conventional methods. Various rapid detection methods have been developed and are generally more sensitive, specific, time-efficient, labour-saving, and reliable than conventional methods. Nucleic acid-based methods such as PCR, mPCR, RT-qPCR, LAMP, NASBA and DNA microarray have high sensitivity and specificity and can overcome the limitations of the culture-based methods, but these methods require trained personnel and specialized instruments. Molecular methods, especially PCR techniques, are very useful tools for diagnosing bacterial infections in routine laboratories, being able to detect clinically relevant antibiotic resistance genes and bacterial isolates growing in biofilms [116,120]. Isothermal amplification techniques, such as LAMP methods, are novel gene amplification methods increasingly used in the specific detection of bacteria. These techniques present some advantages over PCR methods: shorter analysis time, ease of use, inexpensive running costs, higher specificity, and sensitivity, but they have a complicated design, requiring the use of at least four primers and it is also difficult to develop multiplex tests using these methods [117,128].

Numerous biosensor-based methods have recently emerged for the rapid, sensitive, cost-effective, and easy detection of a large number of bacteria [133,134]. The biosensors have two main elements: a bioreceptor and a transducer. The bioreceptor can be a biological material, such as enzymes, antibodies, phages, nucleic acids and cell receptors, or biologically derived material, such as aptamers and recombinant antibodies, or biomimetic imprinted polymers and synthetic catalysts. The transducer can be optical, electrochemical, mass-based, thermometric, micromechanical, or magnetic. Among the most important advantages of these methods are the possibility of miniaturization, portability, as well as the possibility to perform on-site and real-time analyses without the need for complex sample preparation, thus being preferred in routine laboratories for rapid detection of bacteria. The possibility of the use of nanomaterials to increase the sensitivity of the detection method is another advantage of the biosensors. However, the complexity of the matrix can negatively influence the detection of bacteria directly from real samples, the optimization of the sensor being a very important step in its development and requiring a long time [128,133,134,135].

Immunological methods, such as ELISA and ICA, are based on the antigen-antibody specific interaction that leads to a visible reaction in the test medium if the antigen is present in the sample [119]. These methods are widely used for the detection of bacteria due to their advantages, such as short analysis time, ease of use, high specificity, and relatively inexpensive equipment. The advent of commercially available ELISA kits and ICA strips has led to their widespread use for routine testing in some European countries. ICA strips also have the advantage of portability and can be used to detect bacteria in the field. However, these methods have lower sensitivity compared to other methods, such as molecular methods, are sensitiveto temperature and pH changes due to the low antibody stability and work best in the absence of interfering molecules in the samples [116,120,128].

The identification of HAI caused by bacteria can also be achieved based on the detection of specific biomarkers, such as quorum sensing molecules, various virulence factors or other specific metabolites of these bacteria. Usually, the first step in detecting specific toxins or metabolites is the detection of the toxin-producing gene using a molecular method, especially PCR. Unfortunately, this step does not provide convincing evidence of protein expression, so in addition to this genetic analysis, protein analysis must be performed to confirm the presence of the molecule. The most widely used methods for detecting bacterial biomarkers are immunoassays and mass spectrometry-based methods, but recently, due to their advantages, biosensors have gained popularity for detecting these small molecules [127,128,133].

The detection of bacteria involved in HAI is of real importance in the medical field, so researchers are still focused on developing new, improved methods that allow the early detection of infections with these pathogens. Table 6 highlights the main testing methods that are currently used to detect these bacteria.

Classical methods, immunoassays, and biosensors are methods that generally allow only the individual detection of bacteria, so multiple tests are needed to identify infections caused by different pathogens.

The purchase of automated systems to detect a large number of bacteria would make it easier to identify bacterial infections, thus reducing the hospitalization days and associated costs. MALDI-TOF MS is a proteomics-based soft ionization bio-mass spectrometry technique that is currently considered the best method for rapid identification of bacteria from different types of samples, such as blood, environmental or food samples. The three most common MALDI-TOF MS platforms are Bruker Microflex, Vitek MS and Autof MS 1000/2000 [141]. An important advantage of this method is the possibility of detecting bacterial isolates from biofilms. MALDI-TOF MS also allows rapid detection of bacterial resistance, being a promising tool for the identification of bacteria in routine laboratories [128].

Several other systems that allow the rapid identification of different bacteria directly from clinical specimens are commercially available. These systems can be based on PCR methods (Biofire^®^ Filmarray^®^, GeneDisc^®^, BAX^®^System, Cepheid Xpert^®^, EntericBio real-time Gastro Panel I^®^, FilmArray^®^ GI panel, Seeplex^®^ Diarrhea ACE Detection system and xTag^®^ Gastrointestinal Pathogen Panel, which is a FDA-approved and Health Canada-approved panel for the detection of multiple agents of gastroenteritis), nanoparticles-based technology (Verigene^®^ platforms, SeptiFast^®^, SeptiTest^®^) or peptide nucleic acid fluorescent in situ hybridization (OpGen^®^ QuickFISH, Accelerate Pheno system) [20,21,22,23,24,117,141]. Recently, an automated T2Dx instrument platform (T2 Biosystems, Lexington, MD, USA) based on nuclear magnetic resonance technology has been developed that allows the multiplex detection of bacteria from the ESKAPE group in a single whole blood sample without a positive blood culture, in less than 6 h. The detection of bacterial pathogens in other infections, such as respiratory infections, gastroenteritis or sexually transmitted infections isbased on new NAAT technology coupled with different PCR types [156,157].

There are also commercially available systems that allow the detection of bacterial resistance, such as Xpert Carba-R, which is a RT-PCR assay approved for clinical identification of infections caused by drug-resistant organisms, either from pure culture or from rectal swabs, and the FilmArray blood culture identification panel, which is a mPCR that detects a number of bacterial species from positive blood cultures. These systems can identify several beta-lactamases that hydrolyze carbapenem, including NDM, KPC, VIM, OXA-48, and IMP-1, in less than an hour. Another PCR system, the Antibiotic Resistance Genes Microbial DNA qPCR Array can detect beta-lactamase genes as well as fluoroquinolone, macrolide, tetracycline, and aminoglycoside resistance genes, but this system is not approved for clinical use. The Verigene Gram-Negative blood culture nucleic acid test is a microarray-based test that allows the detection of several resistance genes in positive blood cultures within 2 h. Other microarray-based tests, such as the Check-MDR arrays, the CarbDetect assay, and the AMR-ve Genotyping kit, can detect genes associated with resistance to fluoroquinolone, aminoglycosides, macrolides, chloramphenicol, tetracycline, and trimethoprim/sulfamethoxazole, but data on the performance of these tests is limited [115,141,146]. These systems allow the rapid detection of bacteria, favoring the implementation of correct treatment and thus reducing the hospitalization days and additional costs.

## 4. Discussions, Future Trends, and Perspectives

In Romania, as shown by the data from the 2015–2019 reports, HAI have important under-reporting. There are several factors that contribute to these small numbers, including three important factors: the structure of the staff and the hospital, insufficient prevention, and poor identification of HAI. The hospital staff isnot sufficiently educated about HAI, so the management of HAI prevention and treatment can undergo major challenges, with a negative impact on the hospital and the patient. The architectural structure of the hospitals does not always correspond to the current requirements; the hospitals are old buildings, with poor circuits for the patient, with areas with high risk of contamination (crowded wards, small waiting hall, etc.). At the legislative level, there are important recommendations for taking HAI prevention measures, but basic measures, such as hand hygiene, changing gloves after touching patients or surfaces, are often barely followed. Identifying HAI is not a priority, because in Romania, reporting HAI is considered erroneously an inefficient management of infections. As a result, hospitals under-report these events, and the figures are lower than the European average, which gives the false impression that HAI are well controlled. Other underreporting factors are the lack of a multidisciplinary team dealing with HAI, fear of the consequences of reporting, the lack of trained personnel, low staff adherence and compliance with official recommendations, lack of well-equipped laboratories, deficiencies in microbiological diagnosis, and irrational use of antibiotics.

To the above data, two other factors can be added that may contribute to the emergence of antibiotic-resistant strains: the harmful use of antibiotics and the prescription and dispense of antibiotics by the pharmacy. Physicians in both hospitals and clinics tend to prescribe antibiotics more frequently, sometimes without any laboratory results on strain and sensitivity. The law stipulates that antibiotics can only be dispensed with a prescription, but there is a document issued by the Romanian College of Pharmacists that allows pharmacists to release antibiotics, in certain situations, without a prescription. This practice has become widespread, and antibiotics are often released without justifying their need.

In Romania, it is necessary to implement strategies for improving the activities of surveillance, prevention, and limitation of infections in medical units and reducing the causes of the under-reporting of HAI.

To reduce the incidence of HAI there are several measures to prevent or manage these infections. One of the most important and challenging steps is to quickly and correctly identify the bacteria involved in HAI. Traditional methods of identification are slow, which is why rapid methods of identification have emerged. These methods have the major advantage of being faster; however, there is still a period (a few hours) until the results are obtained. To shorten this waiting time, it is necessary to develop even faster diagnosis methods. New rapid methods of identifying bacteria or identifying their resistance positively influence clinical outcomes and are an important part of antibiotic stewardship by reducing both the time to effective treatment and the misuse of antibiotics. Rapid detection methods still have some issues, such as high limits of detection, low sensitivity, high costs, the need for trained personnel, gaps in correlation with the diagnosis gold standards, risk of contamination and clinical validation. Because any new technology can be costly, there is a need for proven cost-effectiveness of these platforms to implement them in the clinical laboratory. An optimal diagnosis method should have some important features: a non-invasive method, portable and miniaturized equipment, need for small quantities of sample, analysis of the sample without prior processing, sensitivity and specificity, easy use, fast results (few minutes), low cost and, finally, the possibility of disinfection of the device, to prevent the risk of transmitting pathogens from one person to another. At the same time, more studies are needed to demonstrate the usefulness of these rapid diagnosis methods on well-established patient segments. Despite these limitations, the future of the field of rapid detection of bacteria is highly promising.

Taking into consideration the complex nature of the HAI, an innovative solution in the diagnosis of HAI could come from the artificial intelligence (AI), which has the capability to extract relevant information (features, patterns, or knowledge) from complex or big data. Thus, AI could improve the analytical performance of many diagnosis methods, compensating for their lack of selectivity and/or sensitivity, especially in complex biological samples. Using the huge computing power of AI, the root causes of HAI occurrence and spread could be identified [158,159].

## 5. Conclusions

This review defines HAI, characterizes the different types of HAI and emphasizes the differences in occurrence and reporting of HAI, showing that the HAI remain a major problem for healthcare systems, endangering the patients’ health and leading to a huge increase in the costs associated with the infection management.

The statistics and reports presented offer an overview of the current situation regarding the HAI in Europe. Thus, it is estimated that EU accounts annually for more than 4 million patients acquiring HAI, adding more than €7 billion annually to the healthcare costs. In the EU, the most common isolated bacteria responsible for HAI are *E. coli*, *S. aureus* and *Enterococcus* spp. More than 425,000 HAI caused by AMR microorganisms and over 33,000 associated deaths occur every year. The incidence and types of HAI vary from country to country, showing the importance of the development of the medical system in the prevention, identification, and management of HAI.

The review focuses on the HAI in Romania as an example of an Eastern-European country, with discussions regarding the reported levels of HAI and the expected levels resulting from point, local studies, the spread of MDR bacteria and the medical staff and general population’s attitude towards HAI. Even though a national tragedy has made the Romanian society very aware of HAI, and the current Romanian legislation regulates the norms of surveillance, prevention, and limitation of the HAI in healthcare units, the official HAI incidence rate in Romania remains under 1%. This under-reporting is caused by lack of training of the medical staff and low rates of HAI diagnosis. There are limited studies on Romanian HAI, but they suggest a much higher number of HAI in Romania, caused by the architecture of hospitals, which do not meet the current required standards, and the scarcity of well-defined protocols for the prevention and the management of HAI. Another cause that contributes to the increase of HAI is represented by bacteria with high levels of MDR. Thus, the MDR levels for *P. aeruginosa*, *A. baumanii* and *K. pneumoniae*, three of the most common bacteria causing HAI, are first, second and third place in Europe, respectively. The possible solutions for HAI prevention and control are also discussed in this review.

The lack of available methods for rapid identification of the bacteria involved in HAI is a major drawback in combating HAI. The advantages and disadvantages of the arsenal of analytical methods (nucleic acid, biosensors, immunological, and MS-based methods, including commercially available ones) are discussed. The current methods used for the identification of the bacteria most frequently involved in HAI are described in detail.

Bacteria are characterized by high adaptability, so they will most likely continue to represent the most important source of HAI, making their prevention, identification, and management very difficult. Hopefully, the future will bring better capabilities in the identification of HAI and an increased awareness of the importance of the prevention and reporting of the HAI. However, to achieve satisfactory numbers of reports, a more detailed analysis of the reasons for under-reporting, and the implementation of strategies leading to the proposed objectives, are needed.

## Figures and Tables

**Figure 1 jcm-11-03204-f001:**
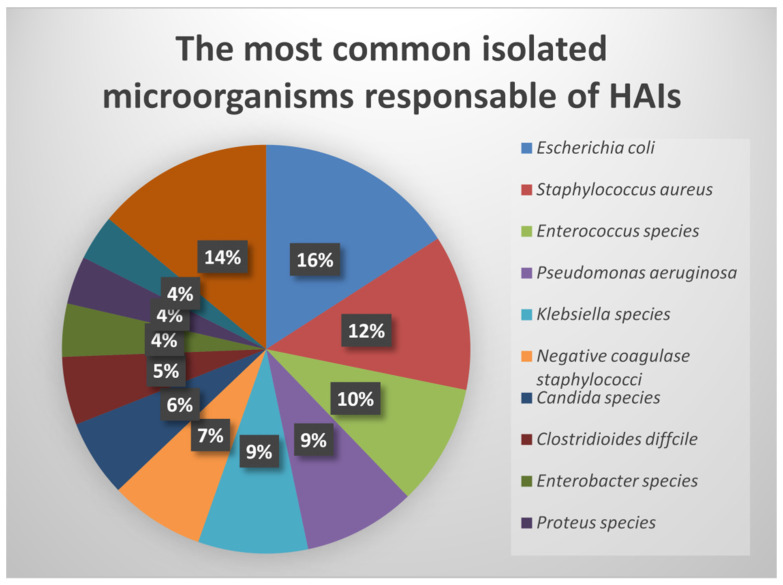
Representation of the most common isolated bacteria responsible for HAI in EU. Adapted from [4].

**Figure 2 jcm-11-03204-f002:**
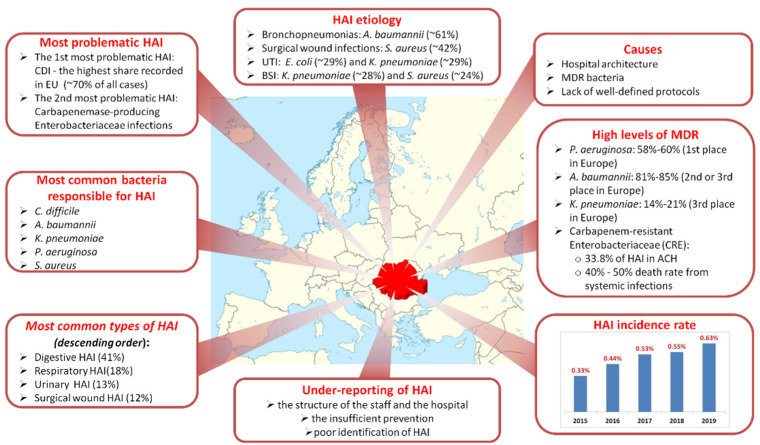
The HAI situation in Romania: The most frequent HAI are digestive and respiratory HAI. The five most common bacteria causing HAI are *C. difficile*, *A. baumannii*, *K. pneumoniae*, *P. aeruginosa* and *S. aureus*. The most problematic HAI is CDI, which has been increasing since 2011. HAI present various etiology depending on the type of HAI. The causes for the high number of HAI include the outdated architecture of hospitals, the bacteria with high levels of MDR (MDR levels for *P. aeruginosa*, *A. baumanii* and *K. pneumoniae* are first, second and third place in Europe, respectively) and the lack of protocols. The HAI incidence rate has increased since 2015, but under-reporting is still observed because of insufficient prevention, identification of HAI and staff training.

**Table 1 jcm-11-03204-t001:** Classification and characterization of HAI according to type.

Characteristics	% of HAI	Causative Organisms	Mortality	Preventable
**Central line-associated bloodstream infections (CLABSI)**				
90% associated with a catheter in the bloodstream [2]	10–15 [38]	*S. aureus* (23%) *Candida* spp. (13%) Coagulase-negative *Staphylococcus* (12%) *Enterococcus* spp. (12%) *Streptococcus* spp. (12%) *E. coli* (8%) *Bacteroides* spp. (6%) [1,2,39]	12–25% [2,39] *P. aeruginosa**:* 50% [40] *A. baumannii:* 29.8–36.9% [41]	65–70% [42]
**Catheter-associated urinary tract infections (CAUTI)**				
Associated with preceding instrumentation or indwelling bladder catheters [1,39]	30–40% [2,43]	*E. coli* *Klebsiella pneumonia*/*oxytoca Enterococcus* spp. *Pseudomonas aeruginosa* *Candida* spp. [1]	13,000 deaths per year in USA [2]	65–70% [42]
**Surgical site infections (SSI). Skin and soft tissue infections (SSTI)**				
Skin, gastrointestinal tract, and female genital tract serve as a reservoir of the healthy flora that may contaminate the surgical site (1) MDR pathogens are increasing [2]	20–24% [2,39,43]	Occasionally are due to airborne spread of skin squames [39] *E. coli* *S. aureus*: most common cause of postsurgical wound infections [30] *Klebsiella* spp. *Enterobacter* spp. *Enterococcus* spp. *Streptococcus* spp. Coagulase-negative *Staphylococcus* [1,2,39]	Over one-third of postoperative deaths [2]	40–60% [2,43]
**Ventilator-associated pneumonia (VAP) and Hospital acquired pneumonia (HAP)**				
5 to 26 % of mechanically ventilated patients develop VAP, after 48 h of intubation [1,2,39] HAP develops after 48 h of admission [1,2] In elderly patients, up to 38% of HAP is due to *S. pneumoniae* [44] MDR pathogens are increasing in VAP [1]	24–27% [2,39]	*S. aureus*: common in HAP [44] and in VAP in intensive care units (ICU) [30] *P. aeruginosa* *Candida* spp. *Klebsiella oxytoca* and *K. pneumoniae* *Streptococcus* spp. *Enterobacter* spp. [1,2]	Up to 50% [2] Pneumococcal pneumonia: 19% [45] *P. aeruginosa:* >30% [46]	55% [42]
**Gastrointestinal infections with *Clostridioides difficile***				
Most common nosocomial cause of diarrhea [47,48] It colonizes the large intestine of approximately 3% of the general population and up to 30–40% of hospitalized patients [48,49,50] Only toxigenic strains, producing toxin A and/or B, are pathogenic [51] CDI risks: use of multiple antibiotics, broader-spectrum agents, and longer duration of therapy [2] Spores infect patients via the fecal-oral route [2]	12% [2]	*Clostridioides difficile*	Up to 7.2% [43] Up to 29,600 deaths per year in the USA [2,52] and up to 3700 deaths per year in Europe [53]	No data

**Table 2 jcm-11-03204-t002:** The World Health Organization (WHO) AWaRe classification of antimicrobials.

Group	Selected Antimicrobials	Characteristics
Access group	Amikacin Amoxicillin Amoxicillin + clavulanic acid Ampicillin Benzathine benzylpenicillin Benzylpenicillin Cefalexin Cefazolin Chloramphenicol Clindamycin Cloxacillin Doxycycline Gentamicin Metronidazole Nitrofurantoin Phenoxymethylpenicillin Procaine benzylpenicillin Spectinomycin Sulfamethoxazole + trimethoprim	This group includes antimicrobials and antimicrobials classes that have activity against a wide range of commonly encountered susceptible pathogens while showing lower resistance potential than antibiotics in the Watch and Reserve groups. Access antimicrobials should be widely available, affordable, and quality assured to improve access and promote appropriate use. Selected Access group antimicrobials (shown here) are included on the WHO Essential Medicine List (EML) as essential first-choice or second-choice empirical treatment options for specific infectious syndromes.
Watch group	Azithromycin Cefixime Cefotaxime Ceftazidime Ceftriaxone Cefuroxime Ciprofloxacin Clarithromycin Meropenem Piperacillin + tazobactam Vancomycin	This group includes antimicrobials and antimicrobials classes that have higher resistance potential. This group includes most of the highest priority agents among the Critically Important Antimicrobials for Human Medicine and/or antimicrobials that are at relatively high risk of selection of bacterial resistance. Watch group antimicrobials should be prioritized as key targets of national and local stewardship programs and monitoring. Selected Watch group antimicrobials (shown here) are included on the WHO EML as essential first-choice or second-choice empirical treatment options for a limited number of specific infectious syndromes.
Reserve group	Azithromycin Cefixime Cefotaxime Ceftazidime Ceftriaxone Cefuroxime Ciprofloxacin Clarithromycin Meropenem Piperacillin + tazobactam Vancomycin	This group includes antimicrobials and antimicrobials classes that should be reserved for treatment of confirmed or suspected infections due to MDRorganisms and treated as “last-resort” options. Their use should be tailored to highly specific patients and settings when all alternatives have failed or are not suitable. They could be protected and prioritized as key targets of national and international stewardship programs, involving monitoring and utilization reporting, to preserve their effectiveness. Selected Reserve group antibiotics (shown here) are included on the WHO EML when they have a favourable risk-benefit profile and proven activity against “Critical Priority” or “High Priority” pathogens identified by the WHO Priority Pathogens List, notablyCRE.

**Table 3 jcm-11-03204-t003:** Excess LOS and estimated costs of HAI worldwide.

Type of HAI	Country/Region	Excess LOS (Days)	Estimated Costs	References
CLABSI	USA	7–15 15 for MRSA	$31,000–65,000 per episode $2.7 billion per year	[12,14,15,39,43]
	Europe	4–14	€4200–13,030 per episode	[61]
	Australia	5.33	$245,371 per 100,000 occupied bed-days	[62]
	Scotland	11.4	£9109 per case	[56]
	Belgium	10.2		[63]
	India	5	$14,818 per case	[61]
	China	12.8		[64]
CAUTI	USA		$13,000 per episode $340 million each year	[2,14]
	France	1.5		[65]
	Belgium	4.6		[63]
	Australia	2–5	$85,081 per 100,000 occupied bed-days	[62,66]
	India	8		[61]
	China	10.3		[64]
SSI	USA	11 23 for MRSA	$3000–29,000 per episode. $43,000 per episode for MRSA. Up to 10 billion per year.	[2,12,14,15,67,68]
	Australia	4–8	$508,243 per 100,000 occupied bed-days	[62]
	Scotland	9.8	£7830 per case	[56,69]
	Belgium	5.9		[63]
	China	11.8		[64]
VAP and HAP	USA	9.1 in the ICU for VAP Up to 9 for HAP	$47,000 per episode for VAP $40,000 per episode for HAP	[2,14,39]
	Australia	2.82	$276,469 per 100,000 occupied bed-days	[62]
	France	6.5		[65]
	Scotland	16.3	£13,024 per case	[56,69]
	India	11		[61]
CDI	USA	3	Up to $17,000 per episode	[14,15]
	Australia	0.5	$8782 per 100,000 occupied bed-days	[62]
	Belgium	12.1		[63]

**Table 4 jcm-11-03204-t004:** Categories of diagnosis tools for bacteria identification.

Characteristics	Advantage	Disadvantage	References
**Nucleic acid-Based Methods**			
Detects specific DNA sequences in the target bacteria Hybridisation of the nucleic acid to a synthetic oligonucleotide Examples: PCR, mPCR, RT-qPCR, LAMP, NASBA, Oligonucleotide DNA Microarray.	Sensitive Specific Faster than culture growth Low detection limit Automated	Require pure samples Prone to contamination Sample processing takes several hours High cost Need for trained personnel Complexity No distinction between viable and not viable bacteria	[120,121]
**Biosensor-Based Methods**			
Can be used for the detection of the whole-cell bacterium, virulence factors, different metabolites, or quorum sensing molecules Examples: electrochemical, optical, piezoelectric biosensors	Low limit of detection Small sample volume Real-time and rapid detection High sensitivity Cost effective method Miniaturization and portability	Sensitive to sample matrix effects	[120,122,123,124]
**Immunological-Based Methods**			
Based on antibody-antigen interactions The sensitivity and specificity are determined by the binding strength of an antibody to its antigen Examples: ELISA, ICA	Sensitive Specific Automated Detection of bacterial toxins	Low sensitivity Prone to false negative results Cross-reactivity with similar antigens Need for specific antibodies High cost	[116,120]
**Mass spectrometry-based** **methods**			
Examples: MALDI TOF MS, HPLC(UPLC)-MS/MS	Very specific and sensitive Rapid and accurate detection Capable of characterizing carbapenemase-producing bacteria Allows the detection of bacterial isolates in biofilms	High acquisition costs for the equipment High annual maintenance costs The need for an extraction or concentration step	[125,126,127]

Abbreviations: PCR, polymerase chain reaction; mPCR, multiplex PCR; RT-qPCR, real-time fluorescence-based quantitative PCR; LAMP, loop-mediated isothermal amplification; NASBA, nucleic acid sequence-based amplification; ELISA, enzyme-linked immunosorbent assay; ICA, immunochromatographic assay; MALDI-TOF-MS, matrix-assisted laser desorption ionization-time of flight; HPLC-MS/MS, tandem mass spectrometry hyphenated to liquid chromatography separation systems.

**Table 5 jcm-11-03204-t005:** Commercially available chromogenic media for the detection of bacteria involved in HAI.

Chromogenic Media	Bacteria	Colour for the Positive Results	Observations	References
**CHROM agar KPC**	*E. coli*	red	Enhanced with mediators that prevent the growth of the carbapenem-sensitive organisms Limitation in the detection of KPC-2-producing *K. pneumoniae* isolates	[130]
	*Klebsiella* spp., *Enterobacter* spp., *Citrobacter* spp.	metallic blue		
	*P. aeruginosa*	translucent cream		
**CHROMagar™ MRSA**	*MRSA*	mauve-colored	Contains a powder base (agar, peptones, yeast extract, salts and chromogenic mix) and a proprietary supplement (powder form qsf 20 L)	[131]
**BBL™ CHROMagar™ MRSA**	*MRSA*	mauve colored	Contains inhibitory agents and cefoxitin	[131]
**CHROMagarTM STEC**	*STEC*	-	Able to detect most of the STEC serotypes	[117]
**CHROMagarTM O157**	O157 STEC	mauve	Is not able to detect most non-O157 STEC	[117]
	Other strains of *E. coli*	blue		
**CHROMagarTM STEC O104**	*O104:H4 STEC*	-	Specifically designed during the 2011 *E. coli* O104:H4 outbreak in Europe	[117]
**ChromID ESBL**	*Enterobacteriaceae*	-	Contains cefpodoxime for the isolation of ESBLs Limitation in detection of OXA-48-like producing isolates that are susceptible to cefpodoxime in the absence of co-production of an ESBL	[130]
**ChromID Pa**	*P. aeruginosa*	purple	Contains ß-alanyl 6 pentylresorufamine	[128]
**ChromID™ MRSA**	*MRSA*	green	Contains α-glucosidase and cefoxitin	[131]
**ChromID VRE**	*E. faecium*	violet	For VRE isolation	[132]
	*E. faecalis*	blue to green		
**ChromID OXA-48**	*CPE*		Direct isolation from clinical samples	[125]
**Brilliance CRE Agar**	*E. coli*	pink	Contains a modified carbapenem for the isolation of CRE	[130]
	*Klebsiella* spp., *Enterobacter* spp., *Citrobacter* spp	blue		
**Brilliance ESBL-agar**	*Enterobacteriaceae*		Detection of ESBLs producing *Enterobacteriaceae*	[130]
**Oxoid Brilliance™ MRSA**	*MRSA*	blue	Result in 18 h Allows the differentiation between MRSA and MSSA	[131]
**Colorex KPC, ID Carba**	*CRE*		Can prevent the growth of Gram-positive and non-carbapenemase producers bacteria	[130]
**Rapidec Carba NP test and Rapid CARB Screen**	*Enterobacteriaceae*		For rapid and efficient detection of KPC, NDM, and OXA-48-like producers	[125]
**MRSASelect**	*MRSA*	strong pink	Contains antimicrobial and antifungal inhibitors	[131]
**InTray Colorex VRE**	*E. faecium* *E. faecalis*	pink to mauve, no differentiation between *E. faecium* and *E. faecalis*	Contain vancomycin For VRE isolation	[132]
**VRESelect**	*E. faecium*	pink		
	*E. faecalis*	blue		
**HardyCHROM VRE**	*E. faecium*	dark blue		
	*E. faecalis*	dark red		
**Spectra VRE**	*E. faecium*	navy blue to pink		
	*E. faecalis*	light blue		

Abbreviations: KPCs, *K. pneumoniae* carbapenemases; OXA-48, oxacillinase-48; ESBL, Extended Spectrum Beta-Lactamase; NDM, New Delhi metallo-β-lactamase; CRE, carbapenem-resistant *Enterobacteriacae*; VRE, vancomycin-resistant Enterococci; MSSA, Meticillin-Sensitive *Staphylococcus aureus*; STEC, Shiga toxin–producing *E. coli*.

**Table 6 jcm-11-03204-t006:** Main testing methods and samples used for the detection of common bacteria involved in HAI.

Bacteria	Testing Methods	Observations	Biological Sample
** *C. difficile* **	TC: isolate the strain on a selective media and detect the toxin production CCCNA: detection of the free toxin directly from the stool sample The test uses Vero cell line to detect the presence of a cytopathic effect neutralized by *C. difficile* antitoxin B. [136]	The diagnosis is complicated because of the phenomenon of asymptomatic carriage of toxigenic *C. difficile* [137] TC and CCCNA are the “gold standards” for *C. difficile* CCCNA is specific, but it requires long time to obtain the results [136]	Liquid or unformed stool
	RT-PCR targeting specific genes: 16S rRNA, the toxin B gene (tcdB), binary toxin genes (cdtA and cdtB), and tcdC gene [136,138]	NAAT are rapid, sensitive Several commercial NAAT are available for direct detection of toxin genes in routine laboratories [136]	
	ELISA and ICA (many commercially available kits) for the detection of GDH and toxins Multi step method: Initial screening for GDH, detects the presence of *C. difficile* [136] Detection of the two free toxins (toxin A–TcdA, toxin B–TcdB) (stool CTA) [136]	ELISA test is rapid, but is not sensitive enough. GDH is a very sensitive assay, but not so specific: it indicates if the bacteria are present, but not if the bacteria are producing toxins [136]	
**MRSA**	Broth enrichment culture prior to inoculation of selective agar media (including chromogenic media)—the standard method in most European laboratories [118]	Speciation of isolates is essential to distinguish *S. aureus* from coagulase-negative staphylococci [139]	Nasal swab Wound swab Skin lesion swab Purulent respiratory secretion (sputum cultures) Blood [140]
	PCR methods that target a DNA segment where the MRSA-specific SCCmec gene meets the *S. aureus* orfX gene [119] GeneXpert MRSA/SA BC Assay targeting the gene encoding staphylococcal protein A (spa), mecA, and the junction region between orfX chromosome and the SCCmec element BD Max StaphSR assay (automated qualitative in vitro diagnostic test) [141] LAMP method: eazyplex^®^ MRSA (a commercial test)	The assays can be performed rapidly, with results available in 1–3 h [118] Allow the differentiation between MSSA and MRSA Eazyplex^®^ MRSA can detect *S. aureus*, *S. epidermidis*, mecA, and mecC from nasal and pharyngeal swabs	
	ICA tests: LAT, based on a monoclonal antibody against a protein produced by the mecA gene (PBP2a) and double gel immunodiffusion/microslide Clearview Exact—a PBP2a-antibody test BinaxNOW *Staphylococcus aureus* Test—does not identify MRSA specifically, but can rule out other staphylococci and Gram-positive cocci [119,127,139]	LAT can distinguish MRSA from MSSA, even in low-level samples Clearview Exact has short analysis time (6 min), similar performances with LAT, but requires fewer steps [119]	
	Aptamer-based sensors, immunosensors for direct detection Fluorescent, phosphorescent and colorimetric biosensors for micrococcal nuclease (MNase) detection [133,134]	Simple and cost effective methods The colorimetric method can be monitored by the naked eye or smartphone camera	
	Toxins detection by HPLC-MS/MS [127]	No isolation of toxin required [127]	
** *S. pneumoniae* **	The routine culture-based identification of *S. pneumoniae* involves bile solubility and optochin susceptibility testing Cultures: inoculation of blood agar (and/or chocolate agar) plates, chromogenic media [140,142]	Pneumococci can be differentiated from other catalase-negative viridans streptococci by their susceptibility to optochin and solubility in bile salts [142]	Blood Urine Cerebrospinal fluid Respiratory secretions (nose, lungs) Sputum
	Molecular methods (PCR, RT-PCR, mPCR, multi locus sequence typing) using an array of pneumococcal specific targets: pneumolysin (ply), autolysin (lytA), pneumococcal surface antigen A (psaA), manganese-dependent superoxide dismutase (sodA) and penicillin binding protein (pbp) [140,143]	The culture-independent method for the detection of pneumococci recommended by the WHO is RT-PCR targeting lytA developed by CDC There is a genetic similarity between *S. pneumoniae* and other closely related species such as *S. mitis*, *S. oralis*, and *S. pseudopneumoniae* [143]	
** *E. coli* **	Culture-based methods (using chromogenic media, Rainbow R Agar O157, MacConkey agar and sorbitol-MacConkey medium for non-sorbitol fermenting *E. coli*) [117,144] Indole testing [145]	There is no culture medium available for the detection of all STEC serotypes [117]	Fecal specimens Enriched stool cultures Blood cultures Urine [117,146]
	Immunoassays that detect Shiga toxin 1 (Stx1) and Shiga toxin 2 (Stx2) antigens Example: Premier R EHEC microwell immunoassay, ProSpectTM Shiga Toxin *E. coli* assay, BioStar R SHIGATOX (optical immunoassay), the Duopath Verotoxin-testTM (ICA test), Premier R EHEC and the Shiga Toxin ChekTM assay (microwell immunoassays) [117,144]	These methods have a prognostic value because there is a clear correlation of Stx2 with the clinical severity of the infection [117] Duopath Verotoxin-testTM differentiates between Stx1-and Stx2-producing STEC	
	Molecular methods, especially mPCR and RT-PCR, and also LAMP, targeting genes encoding Stx1 and Stx2 (stx1 and stx2) and other virulence genes such as the intimin gene, eae, and hemolysin gene, ehx4 [117,144]	Can rapidly detect STEC regardless of the serogroup [144]	
	Aptamer-based biosensors (fluorescent, electrochemical, SERS) [134,135]	Sensitive and specific methods	
** *K. pneumoniae* **	Gram stain Culture-based methods using chromogenic media Phenotypic tests: The carbapenem inactivation method (CIM), Hodge test combined with an EDTA disk test or a disk test using boronic acid compounds [125] Susceptibility testing for carbapenemase producers [147] Indole testing [145]	Can be used for the detection of KPC-producing *K. pneumoniae* isolates Detection of carbapenemase producers based only on minimum inhibitory concentration values may lack sensitivity [147]	Respiratory secretions Blood cultures
	Molecular methods: mPCR, RT-PCR targeting the carbapenemase gene, DNA microarray, LAMP [49,125]	Molecular methods can detect almost all bla genes, including KPCs, NDMs, OXA48-likes, present in bacterial pathogens	
	Electrochemical aptamer-based biosensors [135]		
***Enterobacter* spp.**	Gram stain The gold standard: cultures, with at least two sets of blood cultures, one aerobic and one anaerobic bottle MacConkey agar to determine if the specimen is lactose fermentin Indole testing [145]	*Enterobacter* spp are motile, in contrast to *Klebsiella*, which is not motile [145]	Respiratory secretions Wound swab Skin lesion swab Urine Blood
***Enterococcus* spp. and VRE**	The most common standard methods: MIC determination, disk diffusion, and the breakpoint agar method [148] Culture-based methods: Growth on bile esculin agar and in 6.5% salt broth A positive esculin in combination with a positive PYR reaction to presumptive identification [149] Bile esculin azide agar, triphenyl-tetrazolium chloride azide agar, or an equivalent medium, supplemented with 6 μg/mL of vancomycin, will grow most VRE within 24- to 72-h incubation at 37 °C [150]	*E. faecalis* and *E. faecium* are usually easily identified by most commercial systems With respect to vancomycin intermediate or resistant strains, two key characteristics are motility and pigment *E. casseliflavus* is both motile and possesses a yellow pigment *E. gallinarum* is also motile but non-pigmented *E. faecalis* and *E. faecium* demonstrate neither characteristic [149]	Urine Blood Stool specimens Perirectal cultures Purulent secretions
	PCR-based methods targeting vanA and vanB Commercially available molecular tests: the Roche LightCycler VRE (vanA/vanB) detection 337 assay, the Cepheid Xpert vanA/vanB assay, and the Cepheind Xpert vanA assay [129,148]	Nine different van ligase genes have been described in enterococci, but only the genes encoding vanA and vanB are usually targeted [148]	
** *P. aeruginosa* **	Culture based methods: Gram stain, aerobic incubation on nutrient agar (*Pseudomonas* isolation agar, agars with cetrimide, chromID *Pseudomonas*, acetamide-based agar) [128]	Incubation for 24–48 h at 37 °C on Pseudomonas selective agars in air at 35–37 °C [151]	Exhaled breath condensates Sputum Nasal or throat swab Wound swab Skin lesion swab Skin biopsy Blood Urine [105]
	Molecular methods: PCR (rapid and reliable identification), LAMP and polymerase spiral reaction, targeting several genes (16S rRNA, ecfX, oprL, gyrB, toxA, etc.) [128,151]	Allows the detection of *P. aeruginosa* growing in biofilms	
	Immunoassays: ELISA (commercial kits, such as IgG ELISA kit with three *P. aeruginosa* antigens) and ICA (most often being used the sandwich format that uses gold nanoparticle labeled anti-*P. aeruginosa* monoclonal antibody) and for its QS molecules [128]	Commercial ELISA kits for *P. aeruginosa* are widely used for routine testing in some European countries	
	Biosensors (electrochemical, optical, piezoelectric) for the detection of whole-cell bacterium or for the detection of metabolites and QS molecules [128]	High sensitivity, with low limits of detection	
	MS-based methods: MALDI-TOF-MS and HPLC-MS for the detection of QS molecules and virulence factors (pyocyanin) [128]	Allows the detection of bacterial isolates in biofilms Sample preparation is very important for the detection of bacteria using this method	
***Acinetobacter* spp.**	Preliminary and biochemical tests: Gram stain, catalase test, oxidase test, and hanging drop preparation for motility, CarbAcineto NP test (rapid detection of carbapenemase-producing *Acinetobacter* spp.) [152]	Identification of *Acinetobacter* at species level remains complicated and the traditional phenotypic methods such as culture and biochemical identification are slow, unreliable, and less efficient [153]	Pus Endotracheal aspirate Urine Blood Sputum [154]
	Molecular methods: PCR [152,155]		

Abbreviations: TC, Toxigenic culture; CCCNA, cell culture cytotoxicity neutralization assay; LAT, latex agglutination test; CTA, Cytotoxin assay; GDH, Glutamate dehydrogenase; NAAT, Nucleic acid amplification tests based on real-time PCR; PYR, Pyrrolidonyl-beta-naphthylamide; MIC, minimum inhibitory concentration.

## Data Availability

Not applicable.

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
