# Peer review of "An Overview of Healthcare Associated Infections and Their Detection Methods Caused by Pathogen Bacteria in Romania and Europe"

_jcm, 2022, doi:10.3390/jcm11113204_

Round 1
Reviewer 1 Report
This review reports a broad discussion on nosocomial infections and the increased mortality rates due to the spread of nosocomial microorganisms MDR. It also compares the incidence of the different microorganisms in Europe and the USA, particularly in Romania. Finally, it suggests faster and more sensitive diagnostic methods than older ones, with longer and more difficult processing times. The text is smooth and straightforward. The topic is interesting for the journal's audience.
- The tables could be improved.
- Some references could be integrated (https://doi.org/10.3390/antibiotics10121552) with which to learn more also the trend of antibiotic resistance of these potentially pathogenic species.
Author Response
This review reports a broad discussion on nosocomial infections and the increased mortality rates due to the spread of nosocomial microorganisms MDR. It also compares the incidence of the different microorganisms in Europe and the USA, particularly in Romania. Finally, it suggests faster and more sensitive diagnostic methods than older ones, with longer and more difficult processing times. The text is smooth and straightforward. The topic is interesting for the journal's audience.
We thank the reviewer for the valuable comments and time dedicated to the revision of the manuscript. We modified the manuscript according to the reviewer’s suggestions, which improved the quality of the review.
- The tables could be improved.
The tables have been improved.
- Some references could be integrated (https://doi.org/10.3390/antibiotics10121552) with which to learn more also the trend of antibiotic resistance of these potentially pathogenic species.
The suggested reference was integrated in the text and the following paragraph was added: Enterococcus spp., especially E. faecium and E. faecalis have also received particular attention due to their ability to acquire MDR against many antimicrobial agents used in clinical practice and establish life-threatening infections in patients living with cancer or chronic diseases. In a 5-years study conducted in Salerno, Italy, E. faecium showed high resistance rates against imipenem (86.7%), ampicillin (84.5%), and ampicillin/sulbactam (82.7%), while E. faecalis showed the highest resistance rate against streptomycin (67.7 %) and gentamicin (59.3%) [84].

Reviewer 2 Report
The article deals with an important problem of infections which are connected with providing health services to patients. The aim of the study is well chosen, however, the paper requires a significant revision due to its extensive scope. The authors describe important issues, however, they focus mostly on obvious information about the most common hospital infections, especially on description of infection risk factors, the place and range of their occurrence and pathogens which cause them. According to the reviewer the paper should be shortened and the authors should describe only the situation in Romania. Providing the data from the USA does not comply with the aim of the study nor with its title, and the scope of analyses becomes too extensive.
I recommend a thorough revision of the paper both in terms of content and the subject matter (focusing on important new issues related to HAI prevalence and describing only one country - Romania) as well as citing only current literature and verifying the conclusions of the study which seem to be too general.
Author Response
Reviewer 2
The article deals with an important problem of infections which are connected with providing health services to patients. The aim of the study is well chosen, however, the paper requires a significant revision due to its extensive scope. The authors describe important issues, however, they focus mostly on obvious information about the most common hospital infections, especially on description of infection risk factors, the place and range of their occurrence and pathogens which cause them. According to the reviewer the paper should be shortened and the authors should describe only the situation in Romania. Providing the data from the USA does not comply with the aim of the study nor with its title, and the scope of analyses becomes too extensive.
We thank the reviewer for the valuable comments and time dedicated to the revision of the manuscript. We partially modified the manuscript according to the reviewer’s suggestions, which improved the quality of the review.
We better explained in the introduction part the inter-connection of all the chapters of the review. The review sets the frame about HAI, by presenting the worldwide statistics and the main pathogens responsible for HAI. This information is presented in order to put into context the official reporting of HAI in Romania, which suggests an under-reporting of HAI in Romania. The isolated studies from Romania focused on the main pathogens responsible for HAI, but they don’t offer a complete image. We tried to offer explanation for the lack of reporting in Romania and the chapter dedicated to bacterial detection methods summarized the possible options for rapid diagnosis of bacterial infections.
I recommend a thorough revision of the paper both in terms of content and the subject matter (focusing on important new issues related to HAI prevalence and describing only one country - Romania) as well as citing only current literature and verifying the conclusions of the study which seem to be too general.
The conclusions have been modified to be more specific.

Reviewer 3 Report
Healthcare associated infections is an essential issue in clinic. Therefore, the authors do a well review focusing on the current situation of healthcare-associated infections in Europe and Romania, and regarding strategy.
The authors concluded a well-designed Table about “Excess length of hospital stays and estimated costs of HAI in USA”. If possible, the regions of SSI and associated bacterial distribution are recommend to be mentioned in a similar table.
Also associated treatment strategies for different types of HAI are recommended to supply, which may make your recent review more comprehensive.
Line 146 indicated “It is estimated that hospitals can avoid between 12,000 to 223,000 HAI and save $142 million to $4.25 billion annually with infection prevention measures”. Some explains or examples about infection prevention measures are hope to provide.
In line 164 “HAI prevalence in high-income countries is up to 7.5%, although others have reported rates of 5.7%–7.1%, while in low- and middle-income countries, the prevalence rate ranges between 5.7% and 19.2%”, this sentence gives readers information about different income with different incidence. However, the standards of high, middle or low income are not clear. Please add them.
BSI as an abbreviation term seems lack of full name. please double check it
Figure 2 involved many interesting information. So, I recommend authors to present your essential information as a figure legend below this figure.
Author Response
Reviewer 3
Healthcare associated infections is an essential issue in clinic. Therefore, the authors do a well review focusing on the current situation of healthcare-associated infections in Europe and Romania, and regarding strategy.
We thank the reviewer for the valuable comments and time dedicated to the revision of the manuscript. We modified the manuscript according to the reviewer’s suggestions, which improved the quality of the review.
The authors concluded a well-designed Table about “Excess length of hospital stays and estimated costs of HAI in USA”. If possible, the regions of SSI and associated bacterial distribution are recommend to be mentioned in a similar table.
More information was added to the table and the following paragraph was added to the text: A WHO report found that the SSI are the most investigated and hence most frequent HAI in low- and middle-income countries, with more than 10% (up to 30%) of operated patients usually developing SSI and S. aureus as the most frequent cause of SSI. In these countries, the level of risk for patients undergoing surgical procedures is significantly higher than in developed countries, where SSI rates vary between 1.2% and 5.2%. SSI can prolong hospital stay up to 21 days in settings with limited resources, adding a burden on the patient suffering and the financial cost [61].
Also associated treatment strategies for different types of HAI are recommended to supply, which may make your recent review more comprehensive.
The comment is appreciated, but since the treatment strategies for different types of HAI is a very complex issue, not being the main purpose of this review, only the general principles regarding the treatment of HAI were presented in the text (the following text and table): According to the recommendations [54] of the International Society for Infectious Diseases, hospitals are encouraged to implement a multidisciplinary team to manage the use of antibiotics through the stewardhisp antibiotic program. An infectious disease physician and a clinical pharmacist with infectious disease training form the core team. Prevention is the base of the management of HAI, but when antimicrobial treatment is necessary, there are three groups of antimicrobials that can be used in the management of infections [55], showed in Table 2, in accordance with the results of the microbiological cultures and the antibiogram and the curative treatment.
Table 2. The WHO AWaRe classification of antimicrobials
Group |
Selected antimicrobials |
Characteristics |
Acces group |
Amikacin Amoxicillin Amoxicillin + clavulanic acid Ampicillin Benzathine benzylpenicillin Benzylpenicillin Cefalexin Cefazolin Chloramphenicol Clindamycin Cloxacillin Doxycycline Gentamicin Metronidazole Nitrofurantoin Phenoxymethylpenicillin Procaine benzylpenicillin Spectinomycin Sulfamethoxazole + trimethoprim |
This group includes antimicrobials and antimicrobials classes that have activity against a wide range of commonly encountered susceptible pathogens while showing lower resistance potential than antibiotics in Watch and Reserve groups.
Access antimicrobials should be widely available, affordable, and quality assured to improve access and promote appropriate use.
Selected Access group antimicrobials (shown here) are included on the WHO Essential Medicine List (EML) as essential first-choice or second-choice empirical treatment options for specific infectious syndromes. |
Watch group |
Azithromycin Cef ixime Cefotaxime Ceftazidime Ceftriaxone Cefuroxime Ciprofloxacin Clarithromycin Meropenem Piperacillin + tazobactam Vancomycin |
This group includes antimicrobials and antimicrobials classes that have higher resistance potential. This group includes most of the highest priority agents among the Critically Important Antimicrobials for Human Medicine and/or antimicrobials that are at relatively high risk of selection of bacterial resistance. Watch group antimicrobials should be prioritized as key targets of national and local stewardship programs and monitoring. Selected Watch group antimicrobials (shown here) are included on the WHO EML as essential first-choice or second-choice empirical treatment options for a limited number of specific infectious syndromes. |
Reserve group |
Azithromycin Cef ixime Cefotaxime Ceftazidime Ceftriaxone Cefuroxime Ciprofloxacin Clarithromycin Meropenem Piperacillin + tazobactam Vancomycin |
This group includes antimicrobials and antimicrobials classes that should be reserved for treatment of confirmed or suspected infections due to multi drug-resistant organisms and treated as “last-resort” options. Their use should be tailored to highly specific patients and settings when all alternatives have failed or are not suitable. They could be protected and prioritized as key targets of national and international stewardship programmes, involving monitoring and utilization reporting, to preserve their effectiveness. Selected Reserve group antibiotics (shown here) are included on the WHO EML when they have a favourable risk-benefit profile and proven activity against “Critical Priority” or “High Priority” pathogens identified by the WHO Priority Pathogens List, notably carbapenem-resistant Enterobacteriaceae. |
Line 146 indicated “It is estimated that hospitals can avoid between 12,000 to 223,000 HAI and save $142 million to $4.25 billion annually with infection prevention measures”. Some explains or examples about infection prevention measures are hope to provide.
This section was revised according to reviewer’s suggestion and this paragraph was added: These prevention measures, according to the Romanian healthcare regulations [7] and the WHO recommendations [18], should include hand hygiene using alcoholic solu-tions or soap and water, use of personal protective equipment, using aseptic and safe practices for injecting, preparing and administering parenteral medicinal products, safe handling of medical equipment or contact with potentially contaminated surfaces, decontamination of medical devices and patient care equipment, respiratory hygiene and cough management, environmental cleaning, healthcare waste management, triage of infectious patients, basic principles of standard and transmission-based precautions.
In line 164 “HAI prevalence in high-income countries is up to 7.5%, although others have reported rates of 5.7%–7.1%, while in low- and middle-income countries, the prevalence rate ranges between 5.7% and 19.2%”, this sentence gives readers information about different income with different incidence. However, the standards of high, middle or low income are not clear. Please add them.
The information was added in the following paragraph:
According to the World Bank, the high-income countries have the gross national income (GNI) per capita of at least $12,476, the upper-middle-income countries have the GNI per capita between $4,038 and $12,475, the lower-middle-income countries have the GNI per capita of $1,026 to $4,035 and the low-income countries have the GNI per capita of $1,025 or less [76].
BSI as an abbreviation term seems lack of full name. please double check it
The text was modified as suggested.
Figure 2 involved many interesting information. So, I recommend authors to present your essential information as a figure legend below this figure.
We thank the reviewer for the suggestion. We added the legend bellow figure 2 and reorganized the information in figure 2.
Figure 2. HAI situation in Romania: The most frequent HAI are the digestive and respiratory HAI; The five most common bacteria causing HAI are C. difficile, A. baumannii, K. pneumoniae, P. aeruginosa and S. aureus; The most problematic HAI is CDI, which has been increasing since 2011; the HAI present various etiology depending on the type of HAI; The causes for the high number of HAI include the outdated architecture of hospitals, the bacteria with high levels of MDR (MDR levels for P. aeruginosa, A. baumanii and K. pneumoniae are 1st, 2nd and 3rd place in Europe, respectively) and the lack of protocols; The HAI incidence rate has increased since 2015, but under-reporting is still observed, because of insufficient prevention, identification of HAI.
